# Lysosomal integral membrane protein-2 (LIMP-2/SCARB2) is involved in lysosomal cholesterol export

Saskia Heybrock[1,16], Kristiina Kanerva [2,3,16], Ying Meng[4,16], Chris Ing [5,6], Anna Liang[5,6], Zi-Jian Xiong[6], Xialian Weng[4], Young Ah Kim[7], Richard Collins[8], William Trimble [6,8,9], Régis Pomès [5,6], Gilbert G. Privé [6,10], Wim Annaert [11], Michael Schwake[12,15], Joerg Heeren [13], Renate Lüllmann-Rauch[14], Sergio Grinstein[6,8], Elina Ikonen [2,3], Paul Saftig[1] & Dante Neculai [4]

The intracellular transport of cholesterol is subject to tight regulation. The structure of the lysosomal integral membrane protein type 2 (LIMP-2, also known as SCARB2) reveals a large cavity that traverses the molecule and resembles the cavity in SR-B1 that mediates lipid transfer. The detection of cholesterol within the LIMP-2 structure and the formation of cholesterol−like inclusions in LIMP-2 knockout mice suggested the possibility that LIMP2 transports cholesterol in lysosomes. We present results of molecular modeling, crosslinking studies, microscale thermophoresis and cell-based assays that support a role of LIMP-2 in cholesterol transport. We show that the cavity in the luminal domain of LIMP-2 can bind and deliver exogenous cholesterol to the lysosomal membrane and later to lipid droplets. Depletion of LIMP-2 alters SREBP-2-mediated cholesterol regulation, as well as LDL-receptor levels. Our data indicate that LIMP-2 operates in parallel with Niemann Pick (NPC)-proteins, mediating a slower mode of lysosomal cholesterol export.

[1] Biochemisches Institut, Christian-Albrechts-Universität Kiel, Kiel, Germany. [2] Faculty of Medicine, Anatomy and Stem Cells and Metabolism Research Program, University of Helsinki, Helsinki, Finland. [3] Minerva Foundation Institute for Medical Research, Helsinki, Finland. [4] Department of Cell Biology, and Department of Pathology Sir Run Run Shaw Hospital, Zhejiang University School of Medicine, Hangzhou, Zhejiang, P.R. China. [5] Program in Molecular Medicine, Research Institute, The Hospital for Sick Children, Toronto, Ontario M5G 0A4, Canada. [6] Department of Biochemistry, University of Toronto, Toronto M5S 1A8, Canada. [7] Department of Chemistry and Biochemistry, Queens College, City University of New York, Flushing, New York, USA. [8] Cell Biology Program, Hospital for Sick Children, Toronto M5G 1X8, Canada. [9] Department of Physiology, University of Toronto, Toronto M5S 1A8, Canada. [10] Princes Margaret Cancer Centre, Toronto, ON, Canada. [11] Laboratory for Membrane Trafficking, VIB-Center for Brain and Disease Research, Leuven, Belgium. [12] Faculty of Chemistry, Biochemistry III, University of Bielefeld, 33615 Bielefeld, Germany. [13] Institut für Biochemie und Molekulare Zellbiologie, Zentrum für Experimentelle Medizin, Universitätsklinikum Hamburg-Eppendorf, Hamburg-Eppendorf, Germany. [14] Institut für Anatomie, Christian-Albrechts-Universität Kiel, Kiel, Germany. [15] Present address: Department of Neurology, Northwestern University Feinberg School of Medicine, Chicago, IL 60611, USA. [16] These authors contributed equally: Saskia Heybrock, Kristiina Kanerva, Ying Meng. Correspondence and requests for materials should be addressed to S.G. (email: sergio.grinstein@sickkids.ca) or to E.I. (email: elina.ikonen@helsinki.fi) or to P.S. (email: psaftig@biochem.uni-kiel.de) or to D.N. (email: dneculai@zju.edu.cn)

Cholesterol dictates the biophysical properties of mammalian membranes and is involved in the pathophysiology of various diseases[1]. Its transport and subcellular distribution are subject to tight regulation[2–4]. Cholesterol is an important membrane component of higher eukaryotes, regulating barrier function, membrane fluidity, integral membrane protein operation, membrane traffic and transmembrane signaling processes[4]. It also plays a major role in vascular diseases, diabetes, cancer and several monogenic disorders[5]. Cholesterol delivery to cells by low-density lipoprotein (LDL) requires endocytosis of the LDL-LDL receptor (LDLR) complex. After reaching late endosomes/lysosomes the cholesteryl esters are hydrolyzed by acid lipase. Because the level of cholesterol in the limiting membrane of lysosomes[6] is rather low, a transport process across this membrane followed by delivery to another membrane is thought to exist. Export of cholesterol from lysosomes has been mainly linked to Niemann Pick Type C1 and 2 proteins[7]. Two proteins NPC1 and NPC2, both equipped with sterol-binding domains, cooperate to remove cholesterol from lysosomes[8]. NPC2 is envisaged to bind cholesterol in the lumen and to directly transfer it to NPC1. NPC1 is then thought to transport cholesterol into and/or across the lysosomal membrane. How cholesterol is subsequently delivered to other membranes, such as the ER, endosomes, mitochondria and the plasma membrane is still largely unknown. One of the most abundant membrane proteins of the lysosome, the lysosomal-associated membrane protein LAMP-2, was suggested to accept cholesterol from NPC2, thereby contributing to the NPC1-driven cholesterol efflux from lysosomes[9,10].

Another abundant and central lysosomal membrane component is LIMP-2. LIMP-2 is the only endomembrane member of the CD36 superfamily of scavenger receptors, which also include the plasmalemmal SR-BI and CD36. The receptors are characterized by two transmembrane domains[11], short cytosolic tails and a large ectodomain. LIMP-2 functions as a phospholipid receptor[12], and as a receptor for enterovirus 71 and coxsackieviruses[13].

LIMP2 is known to deliver the acid hydrolase β-glucocerebrosidase to lysosomes[14]; β-glucocerebrosidase binding is mediated by a helical loop localized in the apex of the luminal domain of LIMP-2[15,16] that extends beyond the glycocalyx. Remarkably, unlike other chaperones that cycle between the Golgi and lysosomes, LIMP-2 resides largely in the lysosomes, suggesting that it serves a second, heretofore unknown function. Accordingly, in addition to the apical helical loop, the LIMP-2 ectodomain features a sizeable cavity that could conceivably mediate an additional, likely transport-related function. Notably, the other members of this family—SR-B1 and CD36, which are expressed on the cell surface—mediate the uptake of fatty acids and lipoprotein-associated lipids. Like LIMP-2, the ectodomain of SR-B1 displays an intramolecular tunnel that is thought to deliver cholesterol from bound lipoproteins to the plasma membrane[15]. A similar cavity in the ectodomain of CD36 has been implicated in steroid delivery in *Drosophila*[17]. That the LIMP-2 ectodomain may play a similar role is suggested by the recent detection of a cholesterol molecule buried inside the hydrophobic tunnel of the purified, crystalline protein[12].

In this study, we provide multiple lines of evidence that LIMP-2 participates in lipid transport from lysosomes by transporting cholesterol (and possibly also other lipid species) through its luminal cavity to the lysosomal membrane.

## Results

### Interaction of LIMP-2 and cholesterol.
It is striking that in LIMP-2-deficient mice, which are characterized by ureter pathology, deafness and peripheral neuropathy[18], we observed intracellular structures resembling cholesterol crystals. Such cholesterol-like inclusions are especially prominent in cell bodies

of Schwann cells of peripheral nerves of LIMP-2 knockout mice (Fig. 1a). Similar structures were previously described in macrophages residing in atherosclerotic lesions[19]. Prompted by the presence of a cavity in the extracellular domain of LIMP-2[12], we performed unrestrained molecular dynamic simulations (MDS) to examine whether and where cholesterol binds preferentially within the putative uptake pathway (the tunnel). Twenty-four simulations were initiated with cholesterol in the bulk solution and fifteen with cholesterol docked within the LIMP-2 cavity. A spatial distribution function of cholesterol computed using the combined dataset depicts cholesterol-binding positions from the mouth of the putative uptake pathway to the membrane interface, as well as the helical bundle face (Fig. 1b). The amino acids residues V214, V268, V277, M337, F339, I376, A379, K381, V415 were found to be lining the cholesterol-binding site within the tunnel of the LIMP-2 ectodomain.

To validate the association of LIMP-2 with cholesterol (and/or other lipids) we examined the interaction of recombinant LIMP-2 ectodomain with lipids using UV-crosslinkable bifunctional derivatives, as well as microscale thermophoresis. Trans-sterol bound readily to LIMP-2, while fatty acids, ceramide and sphingosine bound with lower affinity (Fig. 1c and Supplementary Fig. 1A). In addition, using microscale thermophoresis[20] we detected a direct interaction between purified, soluble His-tagged LIMP-2 protein labeled with Red-tris-NTA dye and cholesterol (Supplementary Fig. 1B; $EC_{50} = 112 \pm 32$ nM).

We next investigated whether LIMP-2 deficiency affects the lysosomal clearance of cholesterol from exogenously added low-density lipoprotein (LDL). To this end, control and LIMP-2-deficient mouse embryonic fibroblasts (MEFs) were starved of lipoproteins for 16 h and subsequently exposed to LDL-cholesterol for 6 h. Lysosomal cholesterol was visualized using filipin, a polyene macrolide antibiotic that selectively binds to unesterified cholesterol. Significantly more cholesterol-containing vesicles were observed in cells lacking LIMP-2, suggesting the involvement of LIMP-2 in lysosomal cholesterol efflux (Fig. 1d, e). Neither LIMP-2 overexpression (Supplementary Fig. 1C) nor its depletion (Supplementary Fig. 1D, Fig. 4a) altered the stability of NPC1. Furthermore, the filipin-positive vesicles are likely to be NPC1-positive, given the co-localization of LIMP-2 with NPC1 or with the lysosomal marker LAMP-2 (Supplementary Fig. 1E). Taken together these findings suggest that LIMP-2 deficiency in MEFs promotes a lysosomal cholesterol accumulation despite the presence of NPC1.

To validate the ability of LIMP-2 to bind and possibly transport cholesterol we made use of NPC1-deficient CHO-M12 cells[21]. Loss of NPC1 in these cells leads to a characteristic intralysosomal cholesterol accumulation, as visualized by filipin staining[22]. Importantly, overexpression of murine LIMP-2 in NPC1-deficient cells reduced the cholesterol load, specifically in those vesicles where the mCherry-tagged LIMP-2 was present (Fig. 1f, g). Together, these in vitro and in vivo experiments suggest a role of LIMP-2 in cholesterol transport.

### Cholesterol translocates through the LIMP-2 ectodomain.
To more readily analyze the mechanism underlying LIMP-2-mediated cholesterol translocation out of the lysosomal lumen, we generated an engineered LIMP-2 construct that was targeted to the plasma membrane. To this end, we created a chimera where the cytoplasmic C-terminal domain of LIMP-2 (aa 457–478) was substituted by that of CD36 (aa 463–472) (Supplementary Fig. 2A; hereafter denoted as LIMP-2.cmr), a related scavenger receptor expressed constitutively at the plasma membrane. Visualization was facilitated by tagging the chimera with GFP. We initially verified that the resulting LIMP-2 chimeric

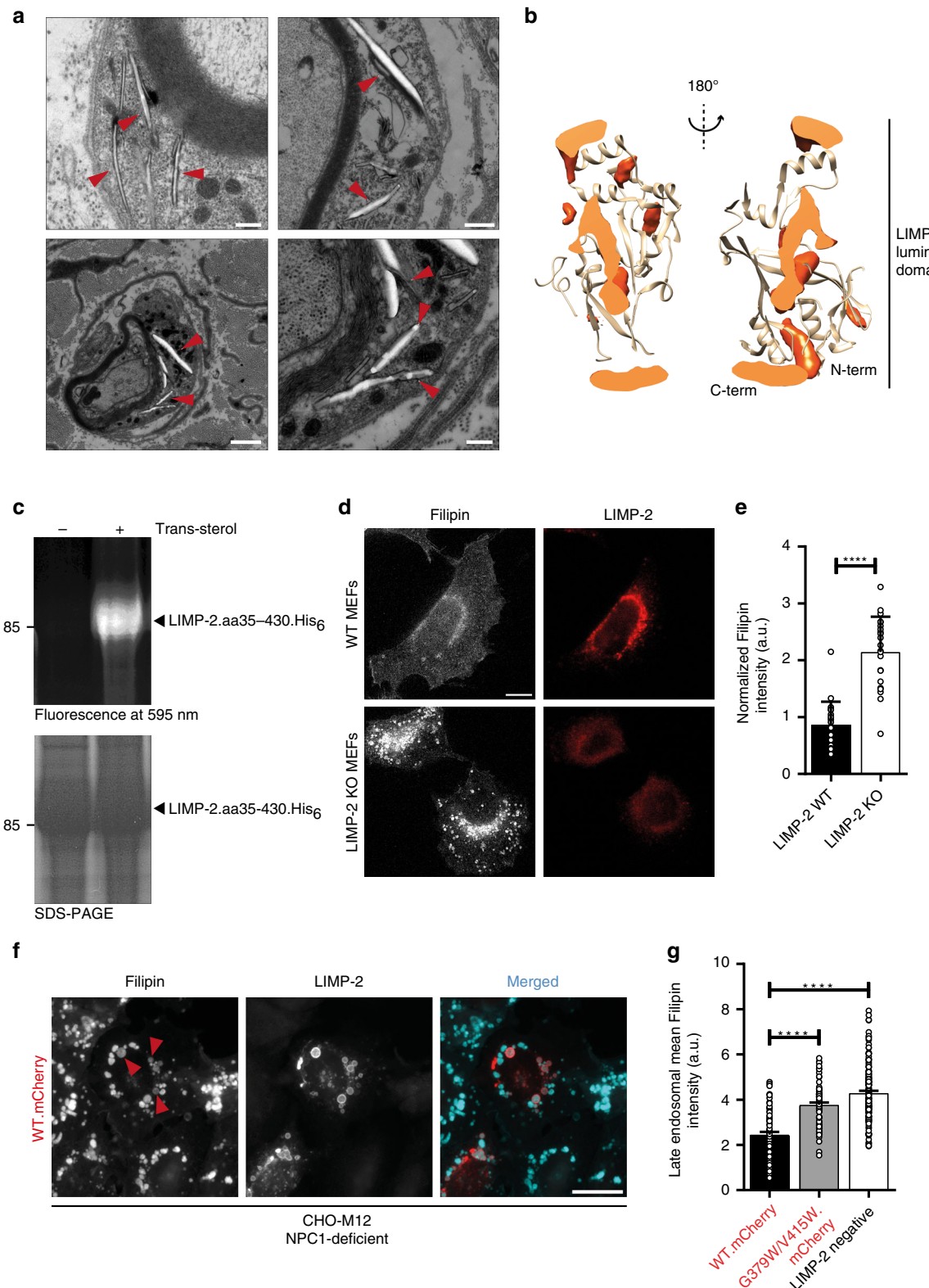

construct was in fact expressed at the plasma membrane. Unlike the full-length wildtype LIMP-2, which is found almost entirely in late endocytic compartments where it colocalizes with internalized dextran, the chimera is clearly discernible at the plasma membrane (Fig. 2a). Exposure of the chimera (but not of full-length LIMP-2) to the extracellular milieu was validated using impermeable biotin derivatives (Fig. 2b).

We proceeded to test the ability of the plasmalemmal LIMP-2. cmr to transport cholesterol. This required a delivery vehicle that would bind to the extracellular (normally luminal) domain of LIMP-2. Given its structural similarity to CD36 and SR-B1, which are known to bind ApoB- and ApoA1-containing lipoproteins, we tested whether the exofacial domain of LIMP-2 would similarly bind these lipoproteins. As shown in Fig. 2c, d, DiI-labeled LDL

**Fig. 1** Physical and functional interaction between LIMP-2 and cholesterol. **a** Electron microscopy revealed characteristic membrane-surrounded cholesterol-like inclusion bodies within the cytoplasm of Schwann cells of the *Nervus phrenicus* of a 6.5 month old (upper panel) and *N. ichiadicus* of a 6.5 month and 23 month old (lower panel) LIMP-2-deficient mouse. Scale bars: (upper panel left, lower panel right) 250 nm (upper panel right) 500 nm, (lower panel left) 1000 nm. **b** Spatial distribution function of cholesterol (orange) in the LIMP-2 extracellular domain (tan) computed from MDS, shown as a cross-section through an axial slice (lighter orange) of the structure. **c** Trans-sterol can be UV-crosslinked to the recombinant ectodomain (luminal part of LIMP-2 (aa35− aa430); +) whereas without the lipid (negative control; −) no crosslinking was observed. The lower panel shows the LIMP-2 protein after Coomassie staining. **d** Wildtype (WT) and LIMP-2-deficient (KO) mouse embryonic fibroblasts (MEFs) were grown for 16 h in lipoprotein-deficient medium and subsequently challenged for 6 h with LDL. Intracellular cholesterol was visualized with filipin. Lysosomes were stained with LIMP-2 in WT MEF cells. Scale bar: 10 μm. **e** Quantification of cellular filipin intensity from D (mean ± SD; LIMP-2-WT $n = 25$, LIMP-2-KO $n = 22$ cells from one experiment; ****$P < 0.0001$, $t$-test). **f** Wide-field fluorescence micrographs of filipin-stained NPC1-deficient CHO M12 cells overexpressing LIMP-2-WT-mCherry. Arrowheads indicate examples of filipin-positive LIMP-2-containing LEs. Note the decrease of filipin fluorescence in LIMP-2-WT-mCherry-containing LEs. Scale bar, 10 μm. **g** Quantification of mean filipin intensity in wild type (WT.mCherry), tunnel-blocking mutant form of LIMP-2 (G379W/V415W.mCherrry) and non-expressing (LIMP-2 neg.) LEs ($n = 75$, 77, 152 organelles, respectively, from 15 cells). Data (mean ± SEM) from two independent experiments, unpaired two-tailed Student's $t$-test (****$P < 0.0001$). Source data are provided as a Source Data file

bound to cells expressing LIMP-2.cmr, but not full-length LIMP-2 (Supplementary Fig. 2B). Similar results were obtained when recombinant LIMP-2 ectodomain tethered on liposomes was tested (Supplementary Fig. 2C). Remarkably, LDL bound to LIMP-2 only at acidic pH, akin to the lysosomal pH that LIMP-2 is normally exposed to (Fig. 2c, d). Like LDL, DiI-labeled HDL also bound to cells expressing LIMP-2.cmr only at acidic pH (Supplementary Fig. 2D). Having identified a suitable delivery vehicle, we tested whether the chimera functioned to deliver cholesterol to the membrane. LDL was chosen as a vehicle because this lipoprotein is internalized and delivered to late endosomes/lysosomes, where LIMP-2 resides. The lipoprotein was doubly labeled with Alexa Fluor 647 succinimidyl ester (to tag the apoprotein) and BODIPY cholesterol (BC; a fluorescent cholesterol analog), and is hereafter denoted as AF647-LDL(BC). After a short incubation with AF647-LDL(BC), CHO and HeLa cells expressing LIMP-2.cmr (tagged with mCherry) displayed fluorescent cholesterol analog in their plasma membrane. Washing the cells repeatedly with HBSS removed the apoprotein, yet fluorescent cholesterol remained associated with the membrane, implying that cholesterol had been delivered from LDL to the plasmalemma (Fig. 2e, Supplementary Fig. 2E). Importantly, we also recapitulated the LIMP-2-mediated transport of LDL-derived cholesterol using recombinant LIMP-2 ectodomain tethered on liposomes (Supplementary Fig. 2F).

Next, we investigated whether the tunnel in the core of the exofacial domain of LIMP-2 is involved in lipid translocation (Fig. 1b). To this end, we mutated amino acids lining the tunnel to bulkier residues (A379W/V415W; Fig. 2f), expected to obstruct lipid translocation. The A379W/V415W double mutation had little effect on the plasmalemmal localization of the chimera (Supplementary Fig. 2G) or on AF647-LDL binding (Supplementary Figs. 2H, I), but significantly reduced its ability to transfer fluorescent cholesterol both in HeLa and CHO cells (Fig. 2g, h and Supplementary Fig. 2J).

To further investigate if the LIMP-2 cholesterol transport activity is restricted to lipoproteins like LDL, we examined whether other types of lipoproteins could deliver cholesterol to LIMP-2. Recently, saposin A (SapA)—a member of the lysosomal saposin family required for the lysosomal breakdown of certain lipids—was shown to form picodiscs with lipids in a manner similar to that of HDL[23]. Like LDL and HDL, SapA-containing lipoproteins not only bound to LIMP-2 but delivered cholesterol to cells in a manner that depended on the integrity of the LIMP-2 tunnel function (Fig. 2i, Supplementary Fig. 2K). Hence, the luminal domain of LIMP-2, and likely its tunnel structure, are critical to deliver lipoprotein-associated cholesterol (and conceivably other lipids) to the membrane.

**Cholesterol delivery to lipid droplets mediated by LIMP-2.** Having demonstrated that its luminal domain is able to transfer cholesterol from the extracellular milieu to the plasma membrane in a tunnel-dependent manner, we turned to in situ (cell-based) assays to assess whether LIMP-2 can similarly transport cholesterol from the lysosomal lumen, to be delivered ultimately to lipid droplets, the storage sites of esterified cholesterol[24]. To follow this pathway in the presence or absence of LIMP-2, human epithelial cells (A431) were labeled for 24 h with oleic acid-conjugated BSA to generate lipid droplets (LD)[25,26]. Cells were subsequently labeled for 2 h with BODIPY cholesteryl linoleate-labeled LDL (BC LN-LDL) that enters late endosomes[27], and chased for up to 4 h in serum-free medium to monitor the transfer of BODIPY cholesterol (BC) from late endosomes to LD (Fig. 3a). BODIPY cholesterol (BC) is the product of BODIPY cholesteryl linoleate (BC LN) hydrolysis by lysosomal acid lipase. The acidic hydrolysis of BC LN is known to be slower than that of native CEs; however this apparent disadvantage makes it a suitable tool to monitor defects in lysosomal cholesterol export by fluorescence microscopy[27]. To determine the role of LIMP-2 in transport across the membrane, cells were treated either with control, NPC1 or LIMP-2 siRNAs (Fig. 3b, c). In control siRNA-treated cells, a considerable fraction of the labeled cholesterol was localized to LD after 4 h, whereas the knockdown of NPC1 resulted in the accumulation of cholesterol in late endosomes, with only minute levels found in LD. Importantly, the downregulation of LIMP-2 also led to an increase in the fraction of BODIPY cholesterol found in late endosomes (Fig. 3b, c). We also used siRNA to compare the contribution of LIMP-2 to that of NPC1. The effect of LIMP-2 silencing was less pronounced than the one of NPC1. The inhibition caused by LIMP-2 knockdown could be prevented by re-expression of murine mCherry tagged LIMP-2, leading to a redistribution of cholesterol similar to that of control siRNA-treated cells (Fig. 3b, c). Importantly, a murine LIMP-2 double mutant (G379W/V415W) corresponding to the human mutations (A379W/V415W) within the tunnel domain described above[14], was less effective in facilitating cholesterol delivery to LD, suggesting a direct involvement of LIMP-2 in this transport process (Fig. 3b, c and Supplementary Fig. 3A). The same tunnel mutant was used to transfect NPC1-deficient CHO M12 cells. In contrast to the transfection with wildtype LIMP-2 (Fig. 1f), the tunnel mutant failed to reduce the vesicular filipin accumulation (Fig. 3d, Fig. 1g).

To gain further insight into the fate of cholesterol, LIMP-2 wildtype and deficient mouse embryonic fibroblasts (MEFs) were incubated with [3H]cholesterol-labeled LDL and the incorporation of the radiolabel into cholesteryl esters (CE) was determined. Thin-layer chromatography (TLC) revealed that the amount of [3H]CE after a 5 h chase was strongly reduced in LIMP-2

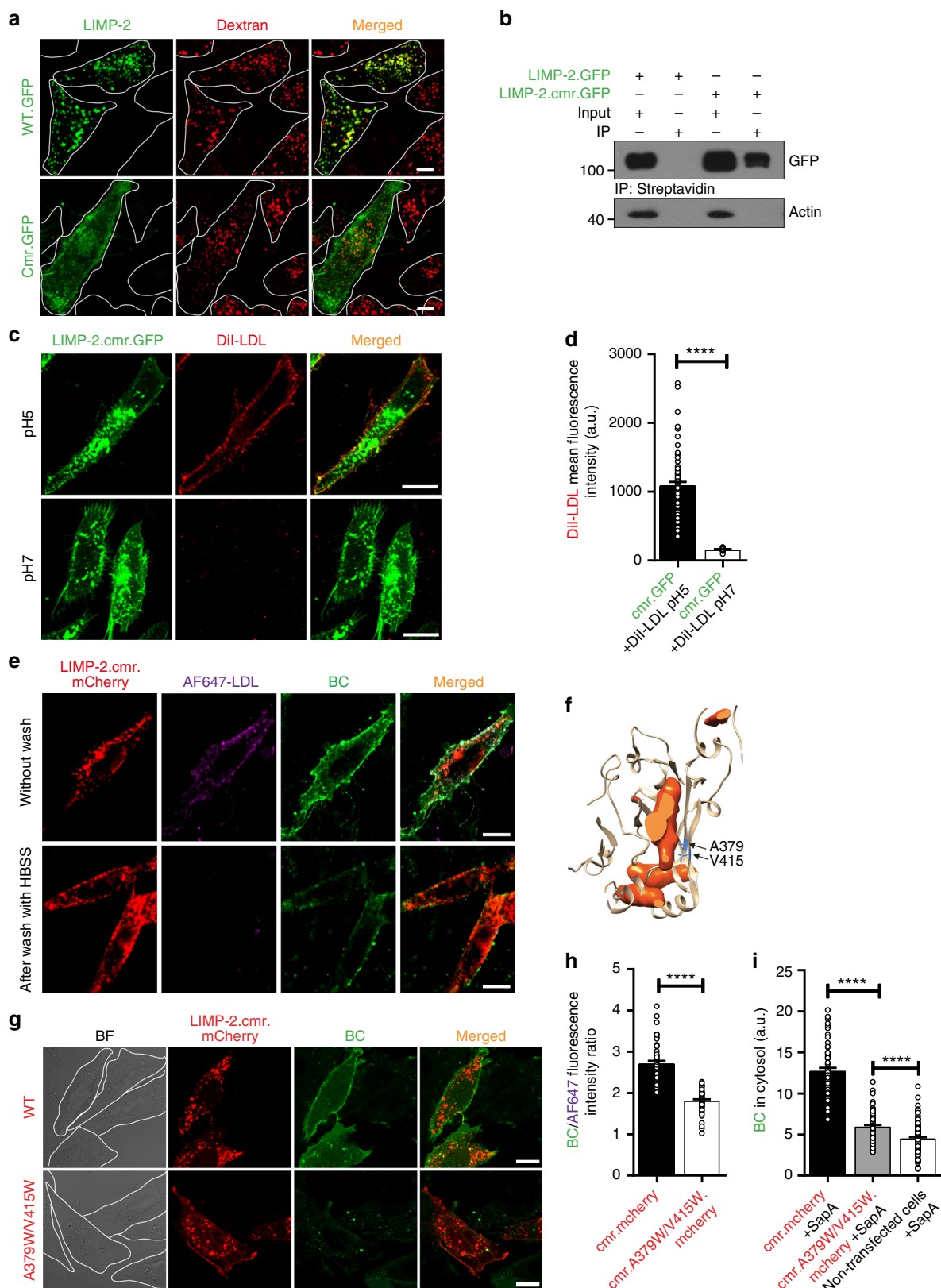

knockout compared to wildtype MEFs (Fig. 3e), in agreement with the imaging results (Fig. 3b, c). Similar results were obtained when cells were radiolabeled with [3H]oleic acid:BSA complex with or without simultaneous addition of LDL, followed by determination of the incorporation of [3H]oleate into CE

(Supplementary Fig. 3B, C). Of note, although the concurrent addition of LDL led to an overall increase in [3H]CE, as expected, the difference between wildtype and LIMP-2 knockout cells was maintained in the absence of LDL (Supplementary Fig. 3C). This further suggests that the cholesterol esterified is not derived solely

**Fig. 2** Lipoprotein-derived cholesterol can translocate through the LIMP-2 ectodomain. **a** Confocal microscopy of HeLa cells transiently expressing C-terminally GFP-tagged wild-type (WT.GFP) or chimeric LIMP-2 (cmr.GFP). Cells were labeled with dextran to visualize late endosomal organelles. Cell boundaries are indicated by dotted lines. **b** Plasmalemmal expression of chimeric C-terminally GFP-tagged LIMP-2 (LIMP-2.cmr.GFP) in CHO cells was detected using Western blotting of biotinylated cell surface proteins after immunoprecipitation using streptavidin-beads. LIMP-2.GFP was used as a negative control. β-actin was used as a loading control for the input samples. **c** Confocal microscopy and quantification (**d**) of DiI-LDL binding to LIMP-2.cmr.GFP in CHO cells. Cells transiently expressing LIMP-2.cmr.GFP were incubated with DiI-LDL at pH 5 (upper panel) and pH 7 (lower panel). pH 5: $n = 78$, pH 7: $n = 9$ cells from two experiments. $t$-test (****$P < 0.0001$). **e** Confocal microscopy of CHO cells transiently expressing LIMP-2.cmr.mCherry and incubated with doubly labeled LDL, AF647-LDL(BC), after binding (upper panel) and washing with HBSS (lower panel). Note that AF647 was removed upon washing but BC was not. **f** Representation of the tunnel-blocking mutations of the human LIMP-2 luminal tunnel, shown as a slice through the body of the protein (sliced solid shown in orange). Residues A379 and V415, which point towards the cavity of the tunnel, are shown in blue. **g, h** Confocal microscopy (**g**) and quantification (**h**) of AF647-LDL(BC)-derived BC transport in CHO cells expressing LIMP-2.cmr.mCherry or LIMP-2.cmr.A379W/V415W.mCherrry proteins. Cells were pre-incubated with doubly labeled LDL (AF647-LDL(BC)) and washed with HBSS. The ratio of green (BC) over far-red (AF647-labeled LDL) fluorescence, measured in ZEN Lite (Zeiss), was used to estimate uptake (**h**); LIMP-2.cmr.mCherry $n = 59$, LIMP-2.cmr.A379W/V415W.mCherrry $n = 50$ cells from 3 experiments. **i** Quantification of BC uptake from SapA(BC) by CHO cells transiently expressing C-terminally mCherry-tagged-wild type (LIMP-2.cmr.mCherry) or tunnel-blocking mutant LIMP-2 chimera (LIMP-2.cmr.A379W/V415W.mCherrry). Also presented is the total fluorescence intensity of non-transfected cells. The total cellular fluorescence intensity of BC was measured using Volocity (LIMP-2.cmr.mCherry $n = 67$, LIMP-2.cmr.A379W/V415W.mCherrry $n = 90$, non-transfected cells $n = 152$ cells from 3 experiments). BC data in **h**, and **i** are the mean ± SEM of triplicate samples (****$P < 0.0001$, unpaired two-tailed Student's $t$-test). a.u., arbitrary units. Scale bars, 10 μm. Source data are provided as a Source Data file

from exogenous sources (i.e., LDL). In both labeling experiments, cells treated with the ACAT inhibitor PKF or with the inhibitor of lysosomal cholesterol efflux U18666A, displayed essentially no radiolabeled CE (Fig. 3e, Supplementary Fig. 3C). These dynamic cell-based experiments underline a physiologically relevant role of LIMP-2 in mediating cholesterol export from lysosomes.

**LIMP-2 is involved in regulating cholesterol homeostasis**. Due to the central role of cholesterol in metabolism, its cellular homeostasis is tightly regulated[1]. Changes in intracellular cholesterol levels are sensed in the endoplasmic reticulum (ER) and cause adaptations in proteins involved in cholesterol synthesis and uptake[1]. Upon cholesterol depletion, a nuclear fragment of the transcription factor sterol regulatory element binding protein 2 (SREBP2) is generated and activates the transcription of sterol-responsive genes like LDLR and HMG-CoA reductase[1,28]. When lysosomal cholesterol egress is hampered, as in the case of NPC1-deficiency, signaling by the nuclear SREBP2 is constitutively active, despite sufficient extracellular supply of cholesterol[29] (Supplementary Fig. 4A). We speculated that depletion of LIMP-2 would result in an altered response of ER-resident regulators of cholesterol homeostasis. To test this possibility, we assessed the effect of depletion or replenishment of cholesterol on the nuclear formation of SREBP2 in the presence or absence of LIMP-2 or NPC1. HeLa cells were treated with control, NPC1 or LIMP-2 siRNAs, followed by 14 h incubation in full medium, cholesterol-depleted medium, or cholesterol-depleted medium with an additional 6 h of re-addition of LDL (Fig. 4a). While re-addition of LDL led to a prominent decrease of nSREBP2 in control cells, no decrease was noted in NPC1 depleted cells. LIMP-2 depletion had a similar, though less drastic effect (Fig. 4a, b). To understand the kinetics of nSREBP2 generation caused by delivery of LDL-derived cholesterol from the lysosome to the ER, we also explored shorter periods (2 and 3 h) of LDL-re-addition (Supplementary Fig. 4, B, C, D). We observed that the knockdown of LIMP-2 delayed the termination of the nSREBP2 signal (Supplementary Fig. 4B). In line with these findings, the expression of the downstream target of nSREBP2, the LDL receptor (LDLR), was increased in the LIMP-2 knockdown cells despite LDL addition. Even though LDL re-addition did not change the level of nSREBP2 in NPC1 deficient cells by 10 h, after 24 h the levels of nSREBP2 were clearly decreased (Fig. 4c, Supplementary Fig. 4E, 4F). This could suggest alternative, though slower export routes of cholesterol from lysosomes, e.g., via LIMP-2. Indeed, the parallel

depletion of NPC1 and LIMP-2 in HeLa cells caused an accumulation of nSREBP2 after 24 h of LDL re-addition (Fig. 4c, d). This effect was even more pronounced when LDLR levels were monitored, i.e., a decreased level of LDLR in NPC1 knockout cells, but no change in LIMP-2/NPC1 double-deficient cells (Fig. 4c, e, Supplementary Fig. 4E–F). An additional downstream target of nSREBP2, 3-hydroxy-3-methyl-glutaryl-coenzyme A reductase (HMGCR), was affected in a similar manner, leading to higher protein levels of HMGCR after re-addition of LDL for 24 h in NPC1/LIMP-2 double-depleted cells than in the cells lacking only NPC1 (Supplementary Fig. 4F, G).

**Discussion**
Unwarranted lipid accumulation in the lysosomes is a hallmark of lipidoses, which can lead to a number of lysosomal storage disorders such as Niemann-Pick disease, Sandhoff disease, Tay-Sachs disease, Farber disease, and Gaucher's disease, depending on the type of accumulated lipid[30]. Whereas the endolysosomal enzymes and nonenzymatic proteins involved in late endosomal/lysosomal lipid metabolism (e.g., the sphingolipid activator proteins (SAPs)) are rather well characterized, not much is known about presumptive late endosome/lysosome lipid transporters, other than NPC1 and LAPTM4B[31]. Although NPC1 polymorphisms were shown to lead to a lysosomal cholesterol accumulation, the lifespan of patients bearing these mutations varied with the age of onset. This, we believe, points to other pathway(s) of lysosomal cholesterol recycling that work in parallel with the NPC1 pathway. These alternate pathways can seemingly transport cholesterol with different kinetics and efficacies, which may also depend on the cell type or total levels of incoming lipoprotein-derived cholesterol.

Herein, we present evidence that LIMP-2 contributes to the recycling of lysosomal LDL-derived cholesterol (and possibly other lipids) alongside the NPC1 pathway. This conclusion is supported by a number of lines of direct experimental (as well as circumstantial) evidence. We demonstrate that LIMP-2 directly interacts with cholesterol and can transport lipoprotein-derived cholesterol or cholesterol analogs to limiting membranes via the LIMP-2 ectodomain tunnel. In addition, knock-down experiments showed that LIMP-2 depletion leads to upregulation not only of cholesterol biosynthesis but also of LDLR expression. Of note, co-precipitation experiments (Supplementary Fig. 4H) did not reveal endogenous interaction of LIMP-2 with NPC1 or

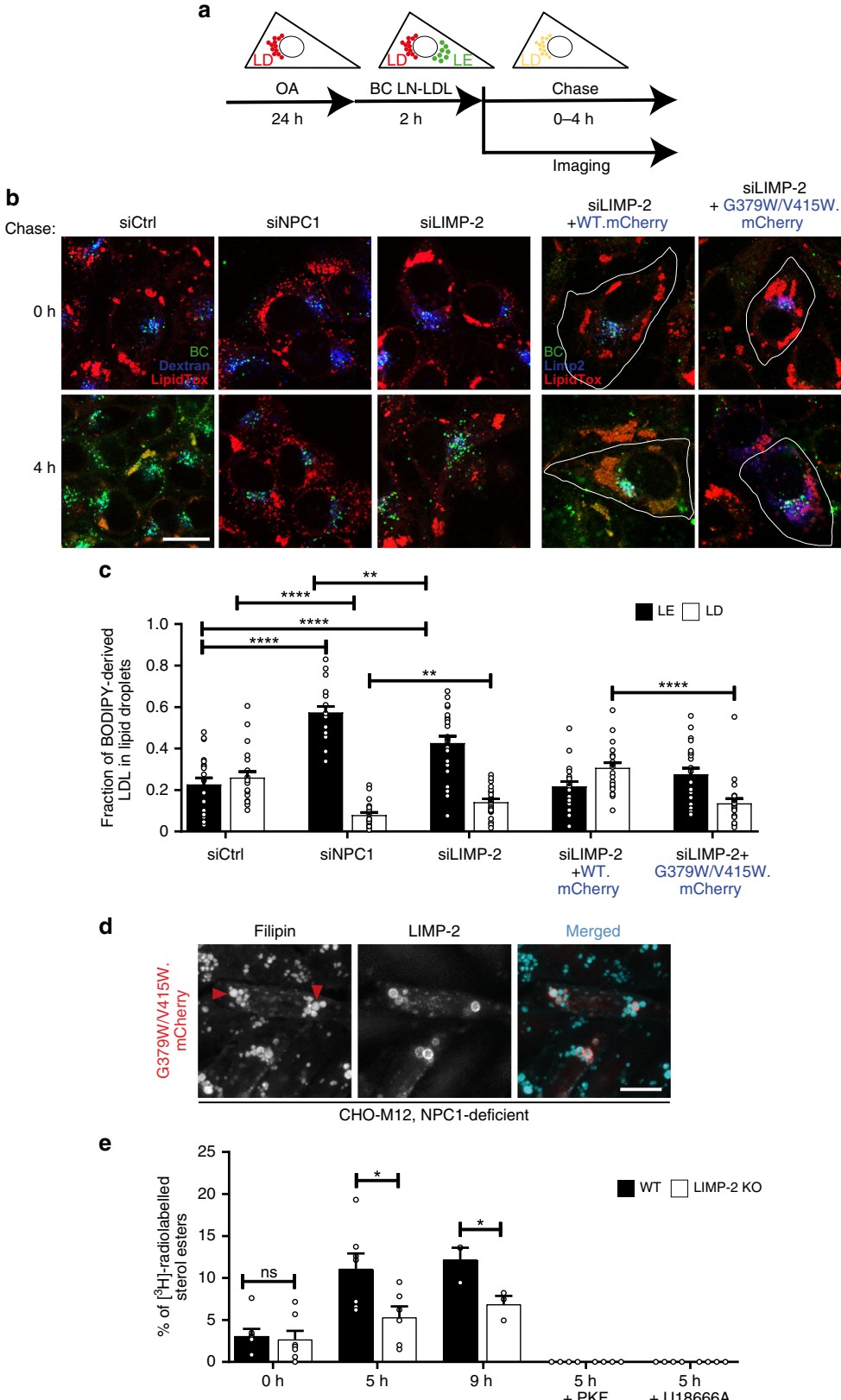

NPC2, arguing that their transport functions are independently regulated.

The cholesterol transporting function of LIMP-2 is not entirely unexpected, considering the cholesterol-binding and fatty acid-binding and uptake capabilities of the other two members of the SR-B family, SR-B1 and CD36. It is well established that SR-B1—an HDL receptor—can selectively transfer HDL-associated cholesterol to the plasma membrane[32]. Therefore, we speculate that endocytosed LDL-associated cholesterol may be handed over to LIMP-2 in a manner resembling the uptake of HDL cholesterol at the membrane by the LIMP-2 homolog SR-B1. A scenario seems likely where a proteolipid complex likely consisting of lysosomal saposins is

**Fig. 3** LIMP-2-WT but not LIMP-2-G379W/V415W overexpression rescues cholesterol efflux from LEs in LIMP-2 deficient cells. **a** Principle of late endosomal BODIPY-cholesterol (BC) efflux assay in A431 cells. Cells are loaded for 24 h with oleic acid-conjugated BSA to generate lipid droplets (LD). In parallel, cells are labeled with dextran to visualize late endosomal organelles (LE). Cells are then labeled for 2 h with BODIPY cholesteryl linoleate-labeled LDL (BC LN-LDL) that enters LE, and chased in serum-free medium to monitor the transfer of BC from LE to LD (labeled with LipidTox during the last 30 min of BC-LN-LDL labeling). **b** Overlaid confocal images of BC (green), dextran/LIMP-2 (blue), and LipidTox (red) in live A431 cells transfected with the indicated siRNAs and/or cDNA constructs (LIMP-2-WT or LIMP-2-G379W/V415W overexpressing cells are outlined) at 0 h and 4 h of chase. Scale bar, 10 μm. **c** Quantification of the fraction of BC fluorescence residing in dextran-positive or LIMP-2-positive late endosomes (LE) vs. LipidTox-positive lipid droplets (LD) after 4 h of chase ($n = 21$ (siLIMP-2, siCtrl), 22 (siLIMP2 + WT.mCherry) or 23 (siNPC1, siLIMP-2 + G379W/V415W.mCherry) cells). Data (mean ± SEM) from 3 experiments, t-test (**$P ≤ 0.01$, ****$P < 0.0001$). **d** Wide-field fluorescence micrographs of filipin stained NPC1-deficient CHO M12 cells overexpressing LIMP-2-G379W/V415W-mCherry. Arrowheads indicate examples of filipin-positive LIMP-2-containing LEs. Scale bar, 10 μm. For quantification see 1 G. **e** Analysis of [$^3$H]cholesterol incorporation into cholesteryl esters in LIMP-2-WT and LIMP-2-KO MEFs in the presence and absence of U18666A or PKF ($n = 7$ (0 h/5 h), $n = 3$ (9 h), $n = 4$ (PKF/U18666A) samples). Data (mean ± SEM) from two (9 h chase) to 3 experiments (5 h chase), unpaired two-tailed Student's t-test (*$P ≤ 0.05$). Source data are provided as a Source Data file

presented to the extended luminal domain of LIMP-2, from where the lipid enters the tunnel and is translocated to the limiting membrane of the lysosome (Fig. 4f). It is also intriguing to speculate that similar to the cholesterol transport mediated at the plasma membrane by the LIMP-2 homolog SR-B1 and cytosolic Aster proteins[33], Aster-like proteins or oxysterol-binding protein related proteins (ORPs) may function downstream of LIMP-2 as cholesterol acceptors. The role of LIMP-2 in such a cholesterol transport route may be of importance for cell types such as Schwann cells. In many other cells the LIMP-2 cholesterol transport may be rather slow compared to the more efficient transport by NPC1, but it may contribute to the variations in the age of disease onset in NPC1 patients. Such variability might be explained by differences in the way cholesterol is offered to the transporter, by the abundance of other ligands transported and/or by differences in the cholesterol derivatives subject to transfer across the tunnel of the LIMP-2 protein. In this context it is interesting that glycosylated cholesterol (GlcChol) was found to be generated by the activity of glycosylceramidase[34], whose transport from the ER to lysosomes depends on the presence of LIMP-2[14].

In summary, our data support a fundamental physiological role of LIMP-2 in lipid transport alongside NPC1 and NPC2. Our results may help direct the development of drugs that can enhance lipid recycling out of the lysosome and/or alleviate the negative consequences of lipidoses.

## Methods

**Materials**. BODIPY-cholesteryl linoleate (BC LN) was synthesized as described[27]. Human LDL was prepared from pooled plasma of four blood donors from the Finnish Red Cross (permit 39/2016) by sequential ultracentrifugation[35] and labeled with BC LN[27]. Lipoprotein-deprived serum (LPDS) was obtained by ultracentrifugation of fetal bovine serum (FBS) as described[35]. Oleic acid/BSA complex (8:1) was generated as previously described[36]. A431 cells were from ATCC (CRL-1555), HeLa wildtype cells (ATCC CCL-2), CHO, CHO-M12 cells[21], mouse embryonic fibroblasts (MEFs)[18] and the NPC1 knockout CRRISPR HeLa cells[37] have been described. Human Silencer Select LIMP-2 siRNAs (cat# 2652) were from Thermo Scientific and human NPC1 siRNAs and control siRNAs against firefly luciferase2; GL2 (both as in ref. [27]) from Sigma.

The following mouse LIMP-2 cDNA constructs were used: pmCherry2-N1-mLIMP-2-mCherry and pmCherry2-N1-mLIMP-2-tunnel mutant (G379W and V415W)-mCherry.

DiI-LDL (20614ES76), DiI-HDL (20611ES76), and LDL (20613ES05) were purchased from YEASEN. Alexa Fluor 647 succinimidyl ester (A20106) was purchased from Life Technologies. Anti-LAMP-2 antibody was purchased from the Developmental Hybridoma Bank (DSHB, University of Iowa, 1:500). Anti-NPC2 (abcam, ab218192, 1:1000 in 5% skim milk), anti-myc (Cell Signaling, #2276, 1:2000 in 5% skim milk), anti-GFP antibody (sc-9996) was purchased from Santa Cruz Biotechnology. Anti-mCherry antibody (M11217) was from ThermoFisher. Anti-actin antibody (M1210-2) and goat anti-mouse IgG HRP (HA1006) were purchased from HuaAn Biotechnology Co., Ltd.

**Cell culture and transfections**. Human epithelial carcinoma A431 cells and MEFs were cultured in DMEM (Lonza), and NPC1-deficient CHO M12 cells in DMEM/F-12 (1:1, Gibco) supplemented with 10% FBS (Gibco), penicillin/streptomycin (100 U/ml each, Lonza) and 2 mM L-glutamine (Gibco).

CHO cells were grown in Ham's F12 medium (Shanghai Basalmedia Technologies Co., Ltd., L410) supplemented with 10% FBS (Gibco, 10270-106), penicillin 100 UI/ml, and streptomycin 100 ug/ml at 37 °C in a 5% CO$_2$ incubator.

siRNAs were transfected in A431 cells with HiPerFect (Qiagen) for 72 h and cDNA constructs with Effectene (Qiagen) for 24 h according to the manufacturer's instructions. Transfections of CHO M12 cells were carried with X-tremeGENE HP DNA transfection reagent (Sigma-Aldrich) for 48 h. Human LIMP-2 plasmids were transiently transfected using Lipofectamin 3000 transfection reagent (Invitrogen, 100022052) or Polyethylenimine (PEI, Polysciences Inc., 23966-2) using the manufacturers' protocols.

**Cloning**. Human LIMP-2 was cloned into pEGFP-N1 (Clontech) and pmCherry-N1 (Clontech) by PCR-driven overlap extension using 2× Phanta Turbo Master Mix (Vazyme,P515-01) and ClonExpress II One Step Cloning Kit (Vazyme, C112-02). hLIMP-2.cmr.GFP, hLIMP-2.cmr.mCherry, and hLIMP-2.cmr.A379V/V415W. mCherry plasmids were generated with a two-step PCR strategy using overlapping primers encompassing the desired mutation and the sequence verified. The N-terminal His6-tagged luminal domain of hLIMP-2 (aa 35–430) was cloned into PB-T-PAF[38] after the protein A tag (protein A-His6-hLIMP-2)[39]. For the generation of mLIMP-2. WT.mCherry and mLIMP-2.G379W/V415W.mcherry, the cDNA of mLIMP-WT and mLIMP-2-G379W/V415W, respectively, were cloned into the vector pmCherry-N1 (Clonetech) by using primers 26 and 27 (see table 'Primers') using Phusion polymerase, 10 mM DNTPs and 5x HF buffer. Restriction digest of insert and vector with EcoRI and BamHI was carried out in 2× Y-Tango buffer for 3 h at 37 °C. Subsequently, the vector was dephosphorylated with FastAP, the samples were separated on a 2% (w/v) agarose gel and purified. Ligation was carried out in a 3:1 ratio over night at 24 °C using T4 ligase and buffer. Chemically competent E. coli XL-1 blue were transformed with 1 μl DNA and plated onto LB plates (+Ampicillin). Clones were picked after overnight incubation at 37 °C, the DNA purified and the constructs analyzed by Sanger sequencing (Eurofins/ GATC). All enzymes and buffers were from Thermo Fisher.

**Primers**

| Name | Sequence |
| --- | --- |
| hLIMP-2.A379W.fw | GAATAATCCTAAAATGGGCCAAGAGGTTCC |
| hLIMP-2.A379W.rv | GGAACCTCTTGGCCCATTTTAGGATTATTC |
| hLIMP-2.his6.rv | gctgctgtgatgatgatgatgatggctgctgcccatGGGCGCGCCctggaaatataaattttC |
| hLIMP-2.his6.fw | cagccatcatcatcatcacagcagcggc TCCGCGTTTAAACTCGAGGT |
| pbtpaf_his_vec.rv | gccctggaaatataaattttc |
| mcherry.fw | gcgggcccgggatccatcgccacc |
| mcherry.rv | ggtggcgatggatcccgggcccgc |
| pmcherry_vec.fw | ggtaccgcgggcccgggatcca |
| pmcherry_vec.rv | GCTTGAGCTCGAGATCTGAGTCCGGTAGCGCTAG |
| LIMP2.456.CD36ct.pEGFPN1.fw | GTTTGGTTTTTACCTGGCTTTATTGTGCATGCAGAT |
| LIMP2.456.CD36ct.pEGFPN1.rv | GATCTGCATGCACAATAAAGCCAGGTAAAAACCAAAC |
| For: 26_SCARB2_EcoRI | actGAATTCCatgggcagatg |
| Rev: 27_SCARB2_BamHI | agtGGATCCGAGGTTCGTATGAG |

**Recombinant protein expression and purification**. Recombinant His-tagged LIMP-2 ectodomain (His$_6$.hLIMP-2(aa35-430)) was expressed in 293 F mammalian cells[38] and purified using standard procedures. The final concentration of the protein solution was 0.4 mg/ml. Saposin A was purified as previously described[23]. SapA was expressed and purified as described by Popovic et al.[23]. Briefly, sapA vector was transformed into Rosetta gami-2(DE3). Cells were grown at 37 °C with 100 mg/ml ampicillin, 34 mg/ml chloramphenicol, 10 mg/ml tetracycline, 50 mg/ml

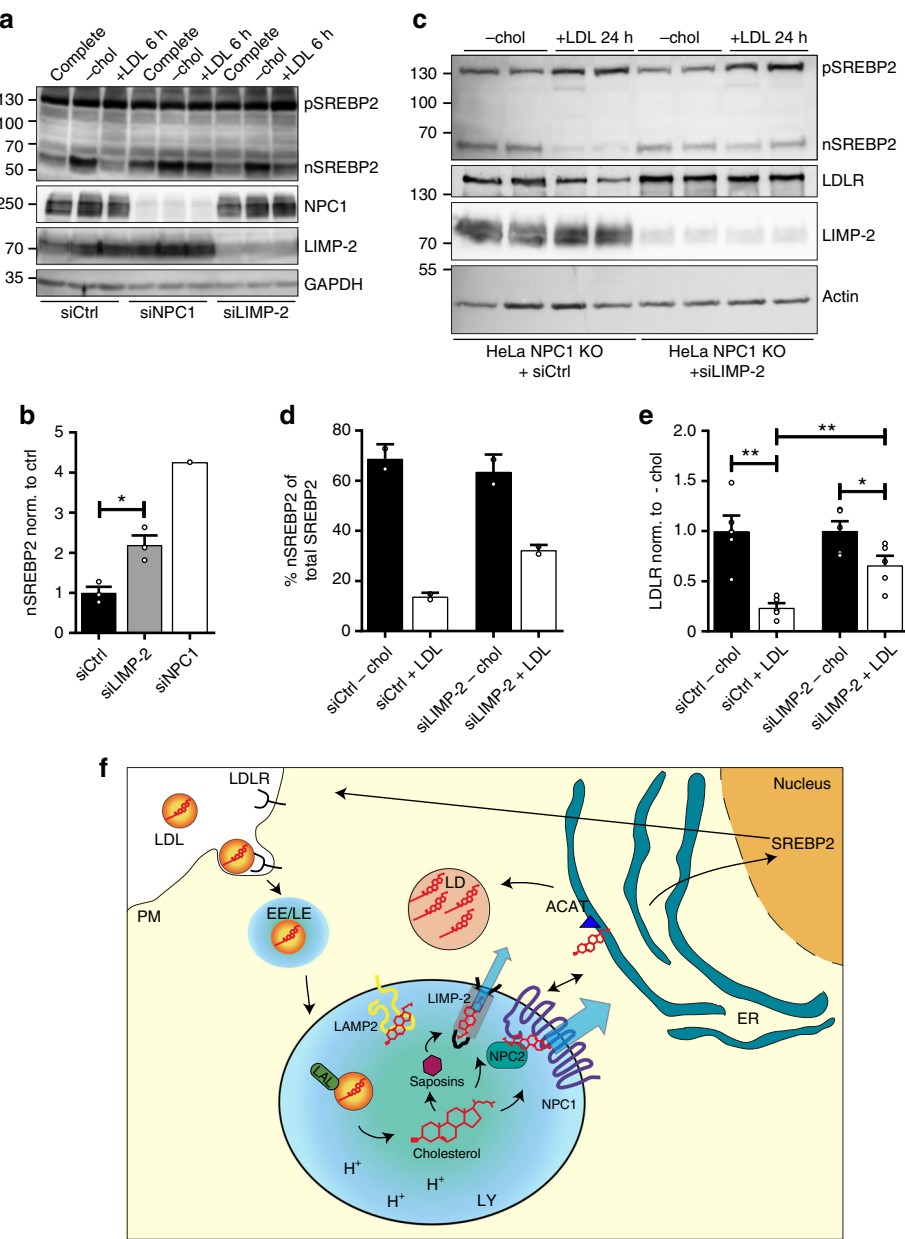

**Fig. 4** Regulation of cholesterol homeostasis is altered in LIMP-2 deficient cells. Analysis of LIMP-2 influence on transcriptional regulation of cholesterol homeostatic genes. **a** Termination of SREBP2 processing in response to abundant cholesterol in the ER (induced by LDL loading) is absent or incomplete in NPC1-silenced and LIMP-2-silenced HeLa cells, respectively. HeLa WT cells were treated with scrambled (ctrl), NPC1 or LIMP-2 siRNA, respectively, and incubated in either full medium, cholesterol-depleted medium or LDL-loaded medium. The processing of SREBP2 in response to high or low cholesterol levels was analyzed by immunoblot. **b** The quantification depicts the amount of nSREBP2 after LDL loading in cells treated with scrambled (ctrl), NPC1 or LIMP-2 siRNA, respectively ($n = 3$; data (mean ± SEM) from two experiments; *$P \leq 0.05$, $t$-test). **c** Analysis of the effect of cholesterol depletion and re-addition of LDL cholesterol for 24 h in NPC1-deficient HeLa cells on nSREBP2 formation. Cells were either treated with control siRNA or with LIMP-2 specific siRNA. The processing of the SREBP2, as well as the expression of the LDL receptor (LDLR) and LIMP-2 was followed by immunoblot. Note that additional depletion of LIMP-2 led to incomplete termination of nSREBP2 fragment and LDLR formation after cholesterol loading for 24 h. **d** Quantification of nSREBP2 levels. Nuclear SREBP2 fragment as percentage of total SREBP2 is depicted. ($n = 2$; data (mean ± SD) from one experiment). **e** Quantification of LDLR levels after cholesterol loading, normalized to the cholesterol-depleted samples ($n = 5$; data (mean ± SEM) from two experiments; *$P \leq 0.05$, **$P \leq 0.01$, unpaired two-tailed Student's $t$-test). **f** Model of the role of LIMP-2 in lysosomal cholesterol efflux. Source data are provided as a Source Data file

streptomycin for 4-6 h and induced with 0.8 mM IPTG overnight at 15 °C and harvested by centrifugation. The pellet was re-suspended in binding buffer (25 mM NaCl, 25 mM Tris-HCl, pH 7.0), lysed by sonication, and centrifuged at 15,000 rpm for 30 min. Then, the supernatant was heated at 90 °C for 15 min. After pelleting down the precipitate by centrifugation at 15000 rpm for 30 min, the resulting supernatant was applied directly to a Q-anion-exchange column (GE Healthcare, 17115301) and the bound protein was eluted with a linear gradient of elution buffer (1 M NaCl, 25 mM Tris-HCl, pH 7.0). The peak fractions containing sapA were collected and further purified by Superdex-75 size-exclusion column (GE

Healthcare, 29-1487-21) in 50 mM Tris-HCl buffer, pH 7.0. SapA fractions were concentrated and stored.

**Microscale thermophoresis (MST)**. MST experiments were performed on a Monolith NT.115 instrument (NanoTemper Technologies). Purified His-tagged LIMP-2 protein, labeled using the Monolith His-Tag Labeling Kit Red-tris-NTA (NanoTemper Technologies) and diluted in buffer with 0.1% Fos-Choline 13 (300 mM NaAc pH 6.0, 150 mM NaCl, 0.05% Tween 20) to a concentration of 25 nM,

was mixed with cholesterol in binding buffer at indicated concentrations ranging from 0.1 nM to 6 uM at room temperature. Fluorescence was determined in a thermal gradient at 60% LED power and medium [low/high] MST power generated by a Monolith NT.115 from NanoTemper Technologies. Data were analyzed by MO.Affinity Analysis v2.3.

**Membrane protein biotinylation and western blot analysis.** The biotinylation assay was performed using the manufacturer's protocol (Pierce Cell Surface Protein Isolation Kit (Thermo Scientific, 89881)). Briefly, 48 h post transfection, CHO cells expressing human LIMP-2 constructs were washed twice with cold PBS and incubated with Sulfo-NHS-SS-Biotin (membrane impermeant) freshly dissolved in cold PBS for 30 min at 4 °C. After quenching of the unreacted succinimidyl ester reagent, cells were lysed and the biotinylated proteins were separated using NeutrAvidin Agarose beads. The immunoprecipitates were analyzed by Western blotting using anti-mCherry (M11217; 1:2.000), anti-GFP(sc-9996; 1:500), and anti-actin (M1210-2; 1:5000) antibodies. β-actin was used to estimate cytosolic contamination in the biotinylation assay. Uncropped western blots are displayed in the Source data file.

**DiI-LDL and DiI-HDL binding assay.** Twenty-four-hours post transfection, CHO cells transiently expressing human LIMP-2 plasmids were washed twice with cold HBSS (Shanghai Basalmedia Technolodies Co. Ltd., B410KJ), pretreated with a NaAc acidic buffer (0.3 M NaCH$_3$COO pH 4.8, 150 mM NaCl, 1 mM MgCl$_2$, 1 mM CaCl$_2$) for 2 min, and incubated with 20 µg/ml DiI-HDL or DiI-LDL resuspended in NaAc acidic buffer for 20 min on ice. Cells were then washed 3 times with NaAc acidic buffer or HBSS and fixed with 4% paraformaldehyde in PBS for 15 min at room temperature.

**Immunocytochemistry and image analysis.** For filipin staining, cells were fixed with 4% paraformaldehyde and quenched with 50 mM NH$_4$Cl followed by incubation with 0.05% filipin (Sigma) in PBS. After washing, the cells were mounted with Mowiol and imaged with a Nikon Eclipse Ti-E inverted wide field microscope equipped with a Nikon motorized stage and an Andor iXon + 885 EMCCD camera with ×40/0.75 NA air objective. The microscope was controlled with NIS-Elements Advanced Research 4.2 software (Nikon).

Late endosomal filipin fluorescence in LIMP-2-mCherry-expressing cells was quantified by ImageJ. The outlines of LIMP-2-positive and LIMP-2-negative LEs (within the same cell) were traced manually. After subtracting background fluorescence, the mean filipin intensity within individual organelles was measured.

Cells in Fig. 2 and S2, except cells in S2F, were visualized using a Zeiss LSM 800 confocal microscope (Zeiss) equipped with an Airyscan detector, and a ×63 oil immersion objective (1.4 NA; Zeiss). Images were acquired and processed with ZEN Blue software (Zeiss). Cells quantified in Fig. 2i and presented in S2F were visualized with a Zeiss ×63 1.4 NA oil-immersion lens using a spinning-disk confocal microscope (Quorum) equipped with a back-thinned EM-CCD camera (C9100-13, Hamamatsu); images were acquired by and processed with Volocity software (Perkin-Elmer).

**Flow cytometry.** To determine binding of AF647-LDL to wild-type and tunnel mutant LIMP-2 chimera proteins, transiently transfected CHO cells were digested with accutase (ThermoFisher, A1110501) and washed twice with cold NaAc acidic buffer (0.3 M NaCH$_3$COO pH 4.8, 150 mM NaCl, 1 mM MgCl$_2$, 1 mM CaCl$_2$). Then, cells were incubated with 20 µg/ml AF647-LDL in acidic buffer for 20 min on ice followed by 3 washes with NaAc acidic buffer and fixed with 2% paraformaldehyde in PBS. Alexa647-LDL binding was quantified using the Median Fluorescence Intensities (MFI) for ligand AF647-LDL for transfected cells from three different experiments using 10,000 cells. The MFIs were the ones reported by the flow cytometry software CytoExpert 2.0. Samples were analyzed with a CytoFLEX low cytometer (BECKMAN) and displayed with CytoExpert 2.0 software.

**Electron microscopy.** For transmission electron microscopy, wildtype and LIMP-2-deficient mice[18] were perfused with 6% glutaraldehyde in phosphate buffer in accordance with the guidelines of the Institutional Animal Care and Use Committee at the University of Kiel. We complied with all relevant ethical regulations for animal testing and research. Tissue blocks were rinsed in phosphate buffer, postfixed in OsO$_4$ for 2 h, and embedded in Araldite or Epon 812 by routine procedures. Ultrathin sections were collected on Ni-grids, contrasted with uranyl acetate and lead citrate, and observed with Zeiss EM 900 or EM 902 microscopes.

**Live-cell BODIPY cholesterol transport assay.** A431 cells, seeded onto glass-bottom dishes (Nunc LabTek 4-well chambered coverglass), were first labeled with 50 µg/ml Alexa Fluor 647-dextran (10,000 MW; Thermo Scientific) supplemented with 200 µM oleic acid/BSA in 5% LPDS to label late endosomal organelles and to induce LDs. The cells were then pulse-labeled for 2 h with 50 µg/ml BC LN-LDL in serum-free DMEM. During the last 30 min of LDL labeling, HCS LipidTox Red or HCS LipidTox Deep Red (1:1000; Thermo Scientific) was added to the medium to label LDs. The cells were then washed and chased in serum-free CO$_2$-independent medium (Gibco) for the indicated times and imaged by live-cell confocal

microscopy. Imaging was performed on a Leica TCS SP8 X attached to a motorized DMi8 inverted microscope with ×63 HC PL APO CS2 water objective (1.20 NA). Experiments were performed at 37 °C in CO$_2$-independent medium (Gibco) supplemented with HCS LipidTox Red/Deep Red (1:1000) in a fully enclosed temperature-controlled environmental chamber. Data were acquired with Leica LAS X (Leica Microsystems). The fraction of BC residing in dextran-positive LEs and LipidTox-positive LDs was quantified from background-subtracted images with ImageJ by using Mander's overlap coefficient as a measure of colocalization.

**[³H]oleic acid incorporation into cholesteryl esters.** To analyze cholesterol esterification, MEFs were first cultured in 5% LPDS for 72 h, washed with PBS and labeled with [³H]oleic acid (5 µCi/ml, Perkin Elmer) in serum-free, 2% defatted BSA (Sigma) medium for 6 h. During labeling, the cells were supplemented with 50 µg/ml LDL, 2 µM U18666A (Sigma) or 2 µM PKF (i.e., Sandoz 58-035, Sigma) as indicated. The cells were collected by scraping in cold 2% NaCl and lipids were extracted by the Blight and Dyer method[38]. Dried lipids were dissolved in chloroform/methanol (9:1), resolved by TLC using hexane/diethyl ether/acetic acid (80:20:1) as the mobile phase and visualized by charring. The cholesteryl ester bands were scraped and radioactivity measured by liquid scintillation counting. The results were corrected for the volume and procedural losses and plotted against the total amount of protein in the sample.

**[³H]cholesterol esterification.** [³H]cholesterol LDL complex was prepared using 50 µCi [1,2-³H(N)]-cholesterol (PerkinElmer) to label 1 mg LDL as previously described (Kanerva et al., 2013). To analyze [³H]-cholesterol incorporation in CEs by esterification, MEFs were cultured in 5% LPDS for 72 h, washed with PBS and incubated with 50 µg/ml [³H]-cholesterol labeled LDL in serum-free medium for 4 h and chased for another 5 h in serum-free medium. During the labeling, the cells were supplemented with 2 µM U18666A (Sigma) or 2 µM PKF (Sandoz 58-035, Sigma) as indicated. The cells were collected and lipids extracted as above. The cholesterol and cholesteryl ester bands were scraped and radioactivity measured by liquid scintillation counting.

**Click chemistry conjugation of LIMP-2.** Fifty-microgram of His-tagged recombinant LIMP-2 ectodomain protein was incubated separately with 50 µM of photoclick lipid probes (photoclick trans-sterol (Avanti, 700147p); pacFA (Avanti, 900401p); pacFA ceramide (Avanti, 900404p); and photoclick sphingosine (Avanti, 900600p)) in the dark for 1 h at 4 °C on a rotating shaker. Next, the samples were exposed to UV light for 15 min (no UV exposure as a negative control) and then mixed with 20 µM rhodamine-azide (ThermoFisher, T10182), 1 mM Tris(2-carboxyethyl)phosphine (TCEP, Sigma-Aldrich), 100 µM Tris[(1-benzyl-1H-1,2,3-triazol-4-yl)methyl]amine (TBTA, Sigma-Aldrich) and 1 mM CuSO$_4$ in TEV buffer (10 mM Tris-HCL pH 8.0, 150 mM NaCl, 1 mM EDTA) at room temperature for 1 h. Afterwards, samples were mixed with SDS sample-loading buffer and loaded without boiling on a 10% SDS-PAGE gel, separated and imaged using a GE Typhoon Phosphorimager.

**Fluorescently labeled lipoproteins.** LDL was labeled with Alexa Fluor 647 succinimidyl ester (ThermoFisher, A20106) according to the manufacturer's protocol followed by extensive dialysis using a microdialysis apparatus for small volumes. Alexa Fluor 647 labeled LDL (AF647-LDL) was next loaded with BODIPY cholesterol (BC) as previously described[40]. Briefly, BC (Avanti, 810255 P), previously solubilized in chloroform, was dried under a stream of argon and was subsequently re-suspended in DMSO. Alexa Fluor 647-labeled LDL was added to the mixture (20 nmol of BC for 1 mg LDL with final DMSO concentration of 10%) and this was further incubated for 2 h at 40 °C, followed by dialysis in PBS containing 1 mM EDTA using a MWCO 3000 dialysis membrane.

**LDL-derived BODIPY-cholesterol transport assay.** Giant liposomes were prepared as previously described[41]. Briefly, liposomes were grown in an Attofluor cell chamber (Thermo Fisher, A7816) where the coverslip was coated with a thin layer of low–melting point agarose which was left out to dry overnight. Fifty-microliter lipid mixture (egg PC (Sigma, 131601C)/cholesterol (Sigma, C8667)/Biotin-PE (Avanti, 860562P)/Rhodamine-PE (Avanti, 810158P)/Egg SM (Avanti, 860061)/DGS-NTA(Ni) (Avanti, 790404C) (25/50/22/0.5/1.5/1) (mol%)) was subsequently soaked up onto the coverslip and dried under argon. Liposomes formed spontaneously upon hydration of the dried lipids with NaAc-based acidic buffer (0.3 M NaCH$_3$COO pH 4.8, 150 mM NaCl, 1 mM MgCl$_2$, 1 mM CaCl$_2$). Liposomes thus formed were transferred to a second Attofluor cell chamber (Thermo Fisher, A7816) in which a coverslip was formerly coated with 7 µg streptavidin in order to immobilize the biotin-PE-containing liposomes to make them amenable for imaging. Next, 40 µg of purified ectodomain of the His-tagged LIMP-2 protein was added to the chamber. After 5 min, Alexa647-labeled LDL lipoprotein was added and LIMP-2 protein and labeled LDL were allowed to interact for 10-min; incubation was followed by imaging using confocal microscopy. For the BODIPY cholesterol transport assay, doubly labeled LDL, AF647-LDL(BC), was incubated with liposomes in the presence or absence of LIMP-2, for 10 min at room temperature. Liposomes were further washed with PBS and visualized by confocal microscopy.

**Preparation of SapA picodiscs**. SapA picodiscs were prepared by a variation of a previously described method[23,42]. Small unilamellar vesicles (SUVs) were prepared by mixing BC-cholesterol, *bis*(monoacylglycero)phosphate (BMP) and egg phosphatidylcholine (DOPC, Avanti Polar Lipids) at a ratio of 10:10:80% moles in chloroform and evaporating the solvent under N2. The dry lipids were dispersed by vortexing in acidic buffer (50 mM sodium acetate pH 4.8, 150 mM NaCl). The suspension was subjected to 10 cycles of freezing (with liquid nitrogen) and thawing followed by sonication in a bath sonicator. SapA was added to a final concentration of 1 mg/mL to solutions of 1 mM liposomes in 50 mM sodium acetate pH 4.8, 150 mM NaCl. The mixture was incubated at room temperature for 20 min and centrifuged at $21,000 \times g$ to pellet unbound liposomes. The supernatant containing the SapA picodiscs was stored at 4 °C.

**BODIPY cholesterol uptake assay**. Twenty-four-hours post transfection, cells were washed twice with warm HBSS and then pretreated with NaAc-based acidic buffer (0.3 M NaCH$_3$COO pH 4.8, 150 mM NaCl, 1 mM MgCl$_2$, 1 mM CaCl$_2$) for 2 min. Then, cells were incubated with of 20 µg/mL BC-containing AlexaFluor647-labeled LDL or 10 µL of the BC-containing SapA lipoproteins (SapA picodisc supernatant prepared as described above) at pH 4.8 for 15 min at 37 °C. Afterwards, cells were washed three times with either acidic buffer or HBSS and fixed in 4% paraformaldehyde for 15 min at room temperature.

**Molecular modeling and simulations**. All-atom molecular models of the LIMP-2 extracellular domain (PDB code: 4F7B, chain A, Neculai et al.[15]) were constructed in solution, in the absence of the N-terminal and C-terminal transmembrane domains or a phospholipid bilayer. Missing side chain heavy atoms were added using MODELLER 9.17[43]. Due to the transmembrane domain truncation in this model, both N-terminal and C-terminal ends of LIMP-2 were modeled as neutral moieties. Sugar molecules were removed at all glycosylation sites and disulfide bonds were formed between residues [C274, C329] and [C312, C318]. All histidine sidechains were singly protonated at Nε. The protein was embedded in a solvated rhombic dodecahedron box consisting of ~150 mM NaCl and ~3 mM cholesterol for a total of ~62,000 atoms (68 Na$^+$, 50 Cl$^-$, 1 cholesterol, 18580 water molecules). The LIMP-2 cavity was initially devoid of water molecules. The protein, cholesterol and water molecules were modeled with the CHARMM36 all-atom force field, CGenFF and TIP3P, respectively[44,45]. Twenty-four initial simulation states were constructed with a cholesterol molecule positioned manually within 10 Å of the protein with diverse initial orientations, and fifteen initial states were initiated with different binding poses of cholesterol within the LIMP-2 cavity. Initial cholesterol binding poses were obtained using AutoDock Vina[46] with six rotatable bonds. Docking was performed with different random seeds in a cubic area of dimensions $50 \times 30 \times 40$ with grid spacing of 0.375 Å centered within the cavity. All simulations were performed with GROMACS 4.6.5[47]. All simulations were conducted at constant temperature (300 K) and pressure (1 atm), using the Nose-Hoover thermostat[48,49] and Parrinello-Rahman barostat[50,51], respectively. Each simulation repeat was independently energy minimized, and subjected to 10 ns of thermalization in the NPT ensemble with protein heavy atoms harmonically restrained in position with a force constant of 1000 kJ/mol nm$^2$. Covalent bonds were constrained with SETTLE[52] and P-LINCS[53] for water and other molecules, respectively. An integration timestep of 2 fs was used. Simulation repeats were conducted for 500 ns and 1000 ns for systems initiated with cholesterol in the solution or docked in the cavity, respectively, for a total of 27 µs of data. Spatial distribution functions were calculated for each simulation repeat on frames extracted at an interval of 1 ns, after backbone alignment of each frame to the LIMP-2 crystallographic structure and removing the initial 50 ns of each repeat. Cholesterol density was averaged over all frames within a fixed set of overlapping spheres that approximated the cavity interior. This analysis was performed using the Visual Molecular Dynamics volmap tool[54] for each repeat, and then averaged over all simulation repeats, weighted by simulation length. Molecular renderings of LIMP-2 with visualization of the density isosurface was performed using[55].

**SREBP2 cleavage assays**. The assay was adapted from[29]. HeLa WT and NPC1 KO cells were transfected with the indicated siRNAs (Silencer Select, Thermo Fisher) for 72 h, then incubated for 16 h in either full medium (DMEM + 10% FCS + 1% Pen-Strep) or cholesterol-depletion medium (DMEM + 5% LPDS + 10 µM mevalonate + 1 µM atorvastatin (Sigma)), respectively. Then, 50 µg/ml LDL was added to the respective samples for 2–24 h at 37 °C. 25 µg/ml calpain inhibitor I (Cayman via Biomol) was added for 1 h at 37 °C before harvesting the cells in PBS + complete protease inhibitor cocktail (Roche). Cells were lysed in PBS + complete protease inhibitor cocktail containing 0.5% Triton-X 100. The lysates were analyzed via SDS-PAGE followed by Western blotting and subsequent immunodetection using anti-SREBP2 (R&D Systems, MAB7119, 1:250 in 5% skim milk), anti-LDLR (abcam, ab52818, 1:1000 in 5% skim milk), anti-HMGCR (abcam, ab1774830, 1:1000 in 5% skim milk), anti-LIMP-2 (self-made, 1:2000 in 5% skim milk), anti-NPC1 (Novus Biologicals, NB400-148, 1:1000 in 5% skim milk), anti-GAPDH (Santa Cruz Biotechnology #sc-25778, 1:5000, 5% skim milk) and anti-Actin (Sigma, A2066, 1:8000 in 5% skim milk) antibodies.

**Statistical analysis**. Data are expressed as mean ± SEM unless otherwise stated. For statistical analysis of the SREBP2 assays, an unpaired student's *t*-test was employed. Changes in protein levels were deemed significant if the *p*-value was <0.05.

**Reporting summary**. Further information on research design is available in the Nature Research Reporting Summary linked to this article.

## Data availability

The data that support the findings of this study are available from the corresponding authors upon reasonable request. A reporting summary for this Article is available as a Supplementary Information file. The source data underlying Fig. 1c, e, g; 2b, d, h, i; 3c, e; 4a, b, c, d, e; Supplementary Figs. 1A, B, C, D; 2G, H; 3B, C; 4B, C, D, E, F,G, H are provided as a Source Data file including also the uncropped and unprocessed scans of western blots.

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

## Acknowledgements

We are grateful to Meryem Senkara for excellent technical help. We would like to thank Suzanne Pfeffer for kindly providing the NPC1-GFP plasmid. We thank for the technical support by the Core Facilities, Zhejiang University School of Medicine. The study was supported by the Deutsche Forschungsgemeinschaft (GRK1459 and FOR2625) to P.S. and S.H., Natural Science Foundation of China (31770938) and the Key Program of Zhejiang Provincial Natural Science Foundation of China (LZ16C050001) grants to D.N., Finnish Cultural Foundation, Häme Regional Fund, and University of Helsinki grants to K.K., Academy of Finland grants 282192, 307415, 312491 to E.I. S.G. was supported by grant FDN-143202 from the Canadian Institutes of Health Research (CIHR). W.A. is supported by VIB, FWO (SBO S006617N) and KU Leuven (C16/15/073)), and SAO-FRA (#20180020).

## Author contributions

S.H., K.K., Y.M., R.P., G.P., A.L., Z.X., R.L. designed and performed experiments, analyzed data, prepared the figures. X.W., Y.K., C.I., R.C., W.T. designed and performed experiments and analyzed data. W.A., J.H. and M.S. provided reagents. S.H., K.K., Y.M. are listed in alphabetical order as equally contributing first authors. S.G., E.I., P.S., D.N. conceived the study, designed experiments, analyzed data, and wrote the paper. All authors have approved the final version of the paper.

## Additional information

**Competing interests:** The authors declare no competing interests.

