## [Peer Review File · Nature Communications]

Reviewers' comments:

Reviewer #1 (Remarks to the Author):

This manuscript examines the role of LIMP-2 in transport of lysosomally-derived cholesterol. The structure of LIMP-2 had previously been determined. The authors use MDS to model the cholesterol binding cavity, and then perform mutagenesis to disrupt cholesterol transport. This is tested by generation of chimeric protein that localizes to the plasma membrane and measuring cholesterol delivery to the ER and down regulation of SREBP-2 target gene expression. The authors conclude the LIMP-2 operates as a lower flux pathway to deliver cholesterol from lysosomes. There are concerns, however, with the experimental systems and methods used in this study, which weaken overall conclusions.

Specific points:

1. It seems likely that there are multiple routes for cholesterol to exit the lysosome. It is clear that cholesterol is able to be transported from the lysosome, albeit slowly, even in the setting of NPC1 or NPC2 LOF. The authors suggest that LIMP-2 may be operating in a lower flux pathway. If that is true, then why would disruption of LIMP-2 trafficking lead to cholesterol accumulation when NPC1 trafficking is intact?
2. In Fig 1, a trans-sterol, a UV-crosslinkable cholesterol probe is used to demonstrate cholesterol binding to the LIMP-2 ectodomain. There are two aspects of the experiment that raise concern. First, excess cholesterol is not used to compete away trans-sterol binding to establish specificity. Second, binding is performed in vitro and not in situ.
3. In Fig 2, the subcellular IF pattern of the WT and A379W/V415W mutant appear to be different. Accordingly, how can one conclude that the reduction in cholesterol transport by the mutant protein is caused by the channel mutation and not an inability to access substrate?
4. The methods used to monitor cholesterol trafficking are suboptimal. Experiments in Fig 3 rely upon use of a BODIPY-cholesterol probe that does not model cholesterol behavior in membrane and cells. Measurement of cholesterol homeostatic responses, such as lipid droplet formation or suppression of SREBP-2 target gene expression are indirect. It is unclear why biochemical methods using radiolabeled cholesterol probes, which permit direct monitoring of the movement of cholesterol cargo, were not used.

Reviewer #2 (Remarks to the Author):

This manuscript reports on a nice, comprehensive set of evidence showing that the lysosomal membrane protein LIMP2 binds lipoproteins at low pH through its luminal domain, that it translocates cholesterol to membranes, and exports free cholesterol from lysosomes. Additional data show that LIMP1 provides an alternate, slower pathway for lysosomal cholesterol export in parallel with the Niemann-Pick disease protein 1 (NPC1).

The study is well designed. The manuscript is clear and well written.

My sole concern is about the interpretation of the effect of the A379W/V415W mutation. This mutant migrates faster than the wild-type in SDS-PAGE (Fig. S2E), suggesting that the lack of activity of this mutant might result from other effects than a block of the luminal tunnel. This shift in the apparent MW should be discussed.

Minor points:

- It would help to move the scheme in Suppl Fig. S3A to the main figure
- Fig 3 legend (filipin staining of the A379W/V415W mutant): "For quantification see 1G", not 1E.

Reviewer #3 (Remarks to the Author):

In this article, the authors provided evidence that LIMP2 delivers free cholesterol from lysosomal lumen to the cytosol. They demonstrated the binding of the luminal domain of LIMP2 to sterols, and show knocking down LIMP2 can cause cholesterol accumulation in lysosomes. However, the data are preliminary and more controls and experiments are required to further strengthen the manuscript. The most critical point of the paper, i.e. LIMP2 can transfer cholesterol directly, is not supported.

1. Figure 1F, there are different classes of endo/lysosomes. Do LIMP2 and NPC1 colocalize on the same set of lysosomes? If not, the rescue data in 1F are not meaningful at all as we are looking at two different populations of endo/lysosomes.
2. Figure 2 in general: why use lipoproteins? If this protein binds sterol directly as proposed in the model of Figure 4, then cholesterol needs to be added as free form. 2E: how is free cholesterol released from LDL? What is the lipase?
3. Figure 2E: after washing, why cherry signal becomes stronger?
4. An in vitro sterol transfer assay using liposomes containing free cholesterol such as DHE needs to be performed to strengthen the conclusion of the paper.
5. Any synergistic effects between NPC1 and LIMP2?
6. In lysosomal lumen, free cholesterol released from LDL is presumably taken up by NPC2. Any interaction between NPC2 and LIMP2?
7. Quality of images for Fig 2 is poor. For Fig2A, bottom panel, chimera blocked the internalization of dextran?
8. Line 235 "...suggest that the cholesterol is not solely derived from exogenous sources...", if this is the case why U18666A treatment resulted in "no detectable cholesteryl ester (line 237)" (Fig 3E).
9. NPC1 blots need to be shown in Fig 4C, Fig S4E, S4F. Did the authors ever mention how they obtained "HeLa NPC1 KO" cells?

Reviewers' comments:

Reviewer #1 (Remarks to the Author):

This manuscript examines the role of LIMP-2 in transport of lysosomally-derived cholesterol. The structure of LIMP-2 had previously been determined. The authors use MDS to model the cholesterol binding cavity, and then perform mutagenesis to disrupt cholesterol transport. This is tested by generation of chimeric protein that localizes to the plasma membrane and measuring cholesterol delivery to the ER and down regulation of SREBP-2 target gene expression. The authors conclude the LIMP-2 operates as a lower flux pathway to deliver cholesterol from lysosomes. There are concerns, however, with the experimental systems and methods used in this study, which weaken overall conclusions.

Specific points:

1. It seems likely that there are multiple routes for cholesterol to exit the lysosome. It is clear that cholesterol is able to be transported from the lysosome, albeit slowly, even in the setting of NPC1 or NPC2 LOF. The authors suggest that LIMP-2 may be operating in a lower flux pathway. If that is true, then why would disruption of LIMP-2 trafficking lead to cholesterol accumulation when NPC1 trafficking is intact?

The reviewer raises an important point. Under basal conditions in the majority of LIMP-2 knockout cells there is no obvious intralysosomal cholesterol accumulation, which indeed suggests that NPC1 compensates for the lack of LIMP-2. However, in specialized cell types –such as Schwann cells– and under conditions of cholesterol overload an accumulation of cholesterol becomes obvious, as evinced by filipin staining (see Fig.1D) and deposition of cholesterol crystals (Fig. 1A).

These findings suggest that LIMP-2 and NPC1 act synergistically and not epistatically. The lack of epistatic interactions between NPC1 and LIMP-2 is supported by the fact that cholesterol is transported out of the lysosomes in settings of NPC1 loss of function (Fig.4C), as well as of LIMP-2 loss of function, which we believe depends critically on the cell type (i.e. Schwann cells). This also agrees with the observation that there was no physical association when NPC1 and LIMP-2 interaction was evaluated by co-immunoprecipitations (Reviewer Figure 3), or by analysis of LIMP-2 and NPC1 co-migration in blue native gel electrophoresis (Reviewer Figure 2; see response to Reviewer 3). In addition, NPC1 protein stability or lysosomal localization was not affected by overexpression of LIMP-2 (Figure S1C, D).

2. In Fig 1, a trans-sterol, a UV-crosslinkable cholesterol probe is used to demonstrate cholesterol binding to the LIMP-2 ectodomain. There are two aspects of the experiment that raise concern. First, excess cholesterol is not used to compete away trans-sterol binding to establish specificity. Second, binding is performed in vitro and not in situ.

We agree with the reviewer's comment regarding the limitations of our UV-crosslinkable cholesterol probe data. Given the inherent issues with competition assays using very poorly soluble lipids, we chose instead to perform microscale

thermophoresis (MST) binding experiments. To this end, binding of cholesterol to the LIMP-2 ectodomain was assessed by MST, using Fos-Choline 13 as delivery vehicle for the cholesterol¹. The results of these experiments unequivocally show that free cholesterol can bind to the luminal domain of LIMP-2 (Figure S1B , EC₅₀= 112 ± 32 nM). Taken together, our *trans*-sterol based experiments, MST assays, and liposome-based assays (Fig. S2F), supported by molecular dynamics simulations (Fig.1B) and the published crystal structure of LIMP-2 (PDB ID: 5UPH) showing a cholesterol molecule bound to the ectodomain, clearly indicate that LIMP-2 can bind cholesterol.

3. In Fig 2, the subcellular IF pattern of the WT and A379W/V415W mutant appear to be different. Accordingly, how can one conclude that the reduction in cholesterol transport by the mutant protein is caused by the channel mutation and not an inability to access substrate?

This experiment refers to a setting where the LIMP-2 WT and mutant proteins were localized to the cell surface, to study cholesterol delivery from LDL to membranes.

Transport of the lipid from the lipoprotein to the plasma membrane involves two steps:

1. binding of the lipoprotein to the LIMP-2 ectodomain followed by
2. transport of the lipid through the LIMP-2 tunnel to the PM.

In the revised manuscript we provide new flow cytometry data (Fig. S2H) that indicate that the binding of LDL to cells expressing the A379W/V415W mutant is comparable to that of the wildtype chimeric LIMP-2 protein. Therefore, we attribute the decrease of the signal (that reflects the incorporation of fluorescent cholesterol into the PM) to the decrease in lipid transport, and not to differential accessibility to the substrate. Furthermore, it is also important to stress that we measured the transport of labeled lipid using the ratio of green signal (fluorescent cholesterol) to the far-red signal (Alexa-647 succinimidyl ester labeled LDL). By calculating the data using this ratiometric approach, which normalizes the transport data per lipoprotein bound, the transport capacity of the scavenger receptors can be estimated independently of their expression levels at the PM. Additionally, we have included new images that better demonstrate the equal expression pattern of the two chimaeras at the cell surface (Fig. 2G, S2I and S2J; and see below Reviewer Figure 1, shown for two different cell types). Importantly, we added an experiment demonstrating equal binding capacity of Alexa-647 succinimidyl ester labeled LDL to both the wildtype and the mutant chimaera (Fig S2H).

Reviewer Figure 1: CHO-K1 and HeLa cells expressing the indicated chimeric constructs revealing comparable surface expression. Scale bar: 10 μ m.

4. The methods used to monitor cholesterol trafficking are suboptimal. Experiments in Fig 3 rely upon use of a BODIPY-cholesterol probe that does not model cholesterol behavior in membrane and cells. Measurement of cholesterol homeostatic responses, such as lipid droplet formation or suppression of SREBP-2 target gene expression are indirect. It is unclear why biochemical methods using radiolabeled cholesterol probes, which permit direct monitoring of the movement of cholesterol cargo, were not used.

We wish to further clarify that there is a clear distinction between the use of fluorescently labeled cholesterol and cholesteryl esters to monitor cholesterol efflux out of the lysosomes. As the majority of cholesterol is transported in the circulation as cholesteryl esters, whether as part of LDL or HDL, we purposely chose to use a fluorescently-tagged cholesteryl ester (BODIPY-tagged cholesterol linoleate) pre-loaded onto LDL to study by fluorescence microscopy the role of LIMP-2 in lysosomal cholesterol efflux. This approach has been successfully used and documented²⁻⁴ and we believe it was the best suited for our purpose as well. Importantly, previously published work shows that despite the slower rate of hydrolysis of BODIPY-tagged

cholesterol linoleate —delivered to endosomes using LDL as a vehicle— defects in lysosomal cholesterol export can be successfully studied¹. We believe that this approach, along with the newly included quantitative assessment of the role of LIMP-2 in lysosome efflux using ³H-cholesteryl ester (new Fig.S3C), fulfills the objective of properly monitoring lysosomal cholesterol efflux.

- 1 Widenmaier, S. B. *et al.* NRF1 Is an ER Membrane Sensor that Is Central to Cholesterol Homeostasis. *Cell* **171**, 1094-+, doi:10.1016/j.cell.2017.10.003 (2017).
- 2 Kanerva, K. *et al.* LDL cholesterol recycles to the plasma membrane via a Rab8a-Myosin5b-actin-dependent membrane transport route. *Dev Cell* **27**, 249-262, doi:10.1016/j.devcel.2013.09.016 (2013).
- 3 Ikonen, E. & Blom, T. Lipoprotein-mediated delivery of BODIPY-labeled sterol and sphingolipid analogs reveals lipid transport mechanisms in mammalian cells. *Chem Phys Lipids* **194**, 29-36, doi:10.1016/j.chemphyslip.2015.08.013 (2016).
- 4 Wustner, D., Lund, F. W., Rohrl, C. & Stangl, H. Potential of BODIPY-cholesterol for analysis of cholesterol transport and diffusion in living cells. *Chem Phys Lipids* **194**, 12-28, doi:10.1016/j.chemphyslip.2015.08.007 (2016).

Reviewer #2 (Remarks to the Author):

This manuscript reports on a nice, comprehensive set of evidence showing that the lysosomal membrane protein LIMP2 binds lipoproteins at low pH through its luminal domain, that it translocates cholesterol to membranes, and exports free cholesterol from lysosomes. Additional data show that LIMP2 provides an alternate, slower pathway for lysosomal cholesterol export in parallel with the Niemann-Pick disease protein 1 (NPC1).

The study is well designed. The manuscript is clear and well written.

My sole concern is about the interpretation of the effect of the A379W/V415W mutation. This mutant migrates faster than the wild-type in SDS-PAGE (Fig. S2E), suggesting that the lack of activity of this mutant might result from other effects than a block of the luminal tunnel. This shift in the apparent MW should be discussed.

We have repeated the surface biotinylation and SDS-PAGE analysis, and found similar migration behavior of the wildtype and mutated chimaera (Figure S2G). We also attach our latest flow cytometry data (Figure S2H) and microscopy analyses (Figure S2I) indicating that LDL binds similarly to cells expressing wildtype LIMP-2 or the mutant LIMP-2 chimaera. We also present better images (see comment and figure above to Reviewer 1) demonstrating comparable surface expression of the wildtype and mutant chimaera.

Minor points:

- It would help to move the scheme in Suppl Fig. S3A to the main figure
- Fig 3 legend (filipin staining of the A379W/V415W mutant): “For quantification see 1G”, not 1E.

We modified the manuscript as suggested, shifting the former Suppl. Fig S3A to the main figures (new Fig. 3A). We also apologize for the oversight, which has been corrected.

Reviewer #3 (Remarks to the Author):

In this article, the authors provided evidence that LIMP2 delivers free cholesterol from lysosomal lumen to the cytosol. They demonstrated the binding of the luminal domain of LIMP2 to sterols, and show knocking down LIMP2 can cause cholesterol accumulation in lysosomes. However, the data are preliminary and more controls and experiments are required to further strengthen the manuscript. The most critical point of the paper, i.e. LIMP2 can transfer cholesterol directly, is not supported.

Below we detail the rationale supporting the concept that LIMP-2 is directly involved in cholesterol transport:

- In our *in vitro* liposome assay, the transport of BODIPY cholesterol from BODIPY cholesterol-preloaded LDL was strictly dependent on the presence of the purified LIMP-2 tethered on the liposomes (Figure S2F).
- This finding was corroborated in assays using cells expressing LIMP-2 chimaera. In these experiments BODIPY cholesterol was transported from BODIPY cholesterol-preloaded LDL to the limiting membrane bilayer in a LIMP-2 chimaera-dependent manner (Figure S2E). Furthermore, transport of BODIPY cholesterol was negligible in cells expressing a tunnel mutant of LIMP-2, despite the fact that binding of LDL to the mutant chimaera was similar to that of the wildtype LIMP-2 chimaera (Figure 2G, S2J).
- These observations lead to the conclusion that the luminal domain must interact with cholesterol, a hypothesis we validated using a number of assays (e.g. MST, fluorescent probes). These results are also in agreement with the finding that LIMP-2 co-crystalizes with cholesterol (PDB ID: 5UPH), and with our molecular dynamics simulations (Fig. 1B).
- The scavenger receptor SR-B1, a close homologue of LIMP-2 that displays 35% sequence identity and 56% similarity, participates directly in cholesterol transport in a non-endocytic manner, and its transport function is dependent on the tunnel present in the SR-B1 extracellular domain (Neculai et al. 2013 Nature).

Neculai D, Schwake M, Ravichandran M, Zunke F, Collins RF, Peters J, Neculai M, Plumb J, Loppnau P, Pizarro JC, Seitova A, Trimble WS, Saftig P, Grinstein S, Dhe-Paganon S. Structure of LIMP-2 provides functional insights with implications for SR-B1 and CD36. *Nature*. 2013 Dec 5;504(7478):172-6.

1. Figure 1F, there are different classes of endo/lysosomes. Do LIMP2 and NPC1 colocalize on the same set of lysosomes? If not, the rescue data in 1F are not meaningful at all as we are looking at two different populations of endo/lysosomes.

In Figure 1F and Figure 3D, we made use of NPC1-deficient CHO-M12 cells to show that lysosomal cholesterol efflux is impaired in cells overexpressing the tunnel-mutant LIMP-2. Obviously, in NPC1-deficient CHO-M12 cells there cannot be co-localization of NPC1 with either over-expressed wildtype or LIMP-2 tunnel mutant. To further address the question of whether different populations of lysosomes are involved, we performed co-localization analyses of NPC1-GFP with LIMP-2 and LAMP-2 in wildtype HeLa cells (Figure S1E), and observed an almost complete colocalization of these markers.

2. Figure 2 in general: why use lipoproteins? If this protein binds sterol directly as proposed in the model of Figure 4, then cholesterol needs to be added as free form. 2E: how is free cholesterol released from LDL? What is the lipase?

We would like to clarify the rationale of these experiments, that seemingly confused the reviewer:

- 1) Cholesterol is poorly soluble in aqueous media and is normally transported in the circulation as a lipoprotein complex; it enters cells via lipoprotein receptor-mediated uptake. If added on its own, cholesterol would be insoluble, but can be rendered soluble by addition of carriers like cyclodextrin. When delivered using a carrier, cholesterol can spontaneously partition into the lipid bilayer in a receptor-independent manner.
- 2) We used LDL that had been preloaded with BODIPY-cholesterol and not BODIPY-cholesteryl ester, which precludes the need for a lipase; in the end-lysosomal compartments, lysosomal acid lipase (LAL) is responsible for the hydrolysis of the cholesteryl esters.

3. Figure 2E: after washing, why cherry signal becomes stronger?

In the original figure two different cells with different expression levels of the LIMP-2 chimaera were shown. We chose to present these images to illustrate the importance of the pH dependence of LDL binding to LIMP-2. Thus, for the washed cells (lower panel), despite the higher expression of the LIMP-2 chimaera in the membrane (higher cherry signal) there was little LDL signal (far red) remaining. However, there was remaining green signal derived from the BODIPY-cholesterol previously incorporated into LDL, which has been transported to the plasma membrane via LIMP-2. This again indicates that LIMP-2 can transport cholesterol directly! Nevertheless, to avoid confusion, we replaced the images in question in the revised manuscript (Fig. 2E).

4. An *in vitro* sterol transfer assay using liposomes containing free cholesterol such as DHE needs to be performed to strengthen the conclusion of the paper.

We followed the reviewer's suggestion and performed new experiments using liposomes. To this end, we used fluorescence microscopy to measure the incorporation of labeled cholesterol (BODIPY-cholesterol) into liposomes in the presence or absence of recombinant LIMP-2, using LDL as a vehicle. Our data directly show that BODIPY-cholesterol preloaded onto LDL can be transported to liposomes *in vitro* (Figure S2F).

5. Any synergistic effects between NPC1 and LIMP2?

See reply to Reviewer #1, point 1. It should also be noted, that LIMP-2 and NPC1 do not co-migrate when analyzed by blue native gel electrophoresis, suggesting that they do not form a stable protein complex (see Reviewer figure 2 below).

Reviewer Figure 2: Blue native gel separation of MEF cell lysates blotted for NPC1 (top panel) and LIMP-2 (lower panel).

6. In lysosomal lumen, free cholesterol released from LDL is presumably taken up by NPC2. Any interaction between NPC2 and LIMP2?

We believe that not all the free cholesterol present in the lysosome is being transported out of the lysosome in an NPC2-dependent manner (see reply to Reviewer #1, point 1.). For instance, our data (Fig. 2I, S2K) indicate that saposin A can solubilize and deliver BODIPY-cholesterol to cells expressing the LIMP-2 chimera. By analogy, we envision that some of the free cholesterol is bound to saposin pico-disks in the lysosome lumen and can then be relayed to and exported via LIMP-2. Experiments testing this hypothesis are ongoing and they will constitute the basis of another manuscript. We also tested whether direct contacts exist between LIMP-2 and NPC2 or between LIMP-2 and NPC1, but failed to detect any convincing interaction (see Reviewer figure 3 below).

Reviewer Figure 3: Immunoprecipitation experiments in HeLa wildtype (WT), LIMP-2 Knockout (KO) and NPC1 Knockout (KO) cells using 2 mg of cell lysate for precipitation (IP) of LIMP-2. No coprecipitation of LIMP-2 with NPC1 or NPC2 could be detected.

7. Quality of images for Fig 2 is poor. For Fig2A, bottom panel, chimaera blocked the internalization of dextran?

In the original manuscript (bottom panel of Fig. 2A) the seemingly extracellular dextran was in fact dextran that had been internalized by endocytosis in an untransfected cell. We have replaced the images in question with higher quality images.

8. Line 235 "...suggest that the cholesterol is not solely derived from exogenous sources...", if this is the case why U18666A treatment resulted in "no detectable cholesteryl ester (line 237)" (Fig 3E).

To address the reviewer's concern, we modified the text as follows:
 "However, the difference between wildtype and LIMP-2 KO cells remained. This suggests that LIMP-2 KO cells have a defect in esterifying both pre-existing membrane cholesterol and LDL-derived cholesterol"

The point we intended to make was that cholesterol esterification is defective in LIMP-2 KO cells both when cholesterol is derived from cell membranes (-LDL) and also when additional cholesterol is brought to the cells via LDL uptake. Overall, cholesterol synthesis in MEFs is slow and the fraction of *de novo* synthesized cholesterol that ends up in cholesteryl esters during the 6-hour ³H-oleic acid labeling period is expected to be negligible.

9. NPC1 blots need to be shown in Fig 4C, Fig S4E, S4F. Did the authors ever mention how they obtained "HeLa NPC1 KO" cells?

Since blots in Fig 4C, S4E and S4F are all from NPC1 KO (and some additional KD of LIMP-2), the lanes corresponding to NPC1 would be negative; for this reason we chose not to show empty blots. Nevertheless, for comparison, we blotted lysates of the exact same cell line, comparing the NPC1 signal of HeLa wildtype and knock-out cells

(Figure S4G). As stated in the manuscript (ref. 36), the NPC1 KO HeLa cells were previously reported and provided by our coauthor Wim Annaert.

- 1 Widenmaier, S. B. *et al.* NRF1 Is an ER Membrane Sensor that Is Central to Cholesterol Homeostasis. *Cell* **171**, 1094-+, doi:10.1016/j.cell.2017.10.003 (2017).

REVIEWERS' COMMENTS:

Reviewer #1 (Remarks to the Author):

The only remaining concern is Point #4. The authors have chosen to measure cholesterol export from the lysosome using flawed (e.g. bodipy-cholesterol) and indirect (LDL-stimulated cholesterol esterification) approaches, which is not optimal because they do not directly measure the trafficking of a faithful cholesterol probe or cholesterol itself from within the lysosome to the ER compartment. That said, the data is consistent with the rest of their findings which alleviates to some extent this concern.

Reviewer #2 (Remarks to the Author):

The revised manuscript and the response to the reviewers satisfactorily addressed all points.

Reviewer #3 (Remarks to the Author):

The authors have addressed my concerns. One aspect the author may include in discussion is the potential cytoplasmic receivers of cholesterol from Limp2. Since Limp2 is similar to SR-B1, a recent paper by Tontonoz and colleagues found ASTER B mediates cholesterol trafficking down stream of SR-B1. Thus, proteins like ORP1, ORP2 or other ASTER B like proteins may function down stream of Limp2.

Final point-to-point reply to the reviewers comments:

Reviewer #1 (Remarks to the Author):

The only remaining concern is Point #4. The authors have chosen to measure cholesterol export from the lysosome using flawed (e.g. bodipy-cholesterol) and indirect (LDL-stimulated cholesterol esterification) approaches, which is not optimal because they do not directly measure the trafficking of a faithful cholesterol probe or cholesterol itself from within the lysosome to the ER compartment. That said, the data is consistent with the rest of their findings which alleviates to some extent this concern.

We were pleased to learn that the reviewer found our data in the current version of the manuscript consistent with our conclusions and that this alleviates his/her concerns. To our knowledge, there is no direct and unambiguous means of assessing cholesterol transfer of cholesterol from lysosomes to the ER in intact cells. Cellular fractionation and chemical determination of the partition of radiolabelled cholesterol between lysosomes and the ER as a function of time is not only very expensive and time-consuming, but is also imperfect, confounded by impurities in the fractions and by redistribution of cholesterol during fractionation. Importantly, as suggested by this reviewer, we did use radiolabeled cholesterol (pre-loaded onto LDL) to measure its biochemical incorporation into cholesterol esters, a reliable measure of its exit from lysosomes (Fig 3D). We appreciate that there is always room for future improvement, but nevertheless believe that a small technical refinement would not alter the conclusions of the manuscript, as acknowledged also by the reviewer.

Reviewer #2 (Remarks to the Author):

The revised manuscript and the response to the reviewers satisfactorily addressed all points.

We are grateful that the reviewer was satisfied with our revision.

Reviewer #3 (Remarks to the Author):

The authors have addressed my concerns. One aspect the author may include in discussion is the potential cytoplasmic receivers of cholesterol from Limp2. Since Limp2 is similar to SR-B1, a recent paper by Tontonoz and colleagues found ASTER B mediates cholesterol trafficking down stream of SR-B1. Thus, proteins like ORP1, ORP2 or other ASTER B like proteins may function down stream of Limp2.

We are grateful that the reviewer found that we addressed all his/her concerns. We have taken up the reviewer's suggestion and included a sentence and the mentioned reference on ASTER B in the discussion section.